# Hand-to-Face Contact of Preschoolers during Indoor Activities in Childcare Facilities in the Republic of Korea

**DOI:** 10.3390/ijerph192013282

**Published:** 2022-10-14

**Authors:** Hyang Soon Oh, Mikyung Ryu

**Affiliations:** 1Department of Nursing, College of Life Science and Natural Resources, Sunchon National University, Suncheon 57922, Korea; 2College of Nursing, Daegu University, Daegu 42400, Korea

**Keywords:** preschool children, self-inoculation, contact behavior, hand hygiene

## Abstract

Purpose: This study aimed to characterize hand-to-face contact (HFC) in children and analyze the factors that affect HFC behaviors of preschoolers in childcare facilities in Korea. Methods: Thirty preschoolers aged between 13 and 84 months were enrolled with parents’ voluntary participation. Videotaping (10 children/childcare center/2 h) and video reading was performed from 23 November 2018 to 7 January 2019. Results: A total of 2719 cases of HFC were observed in 30 participants within 2 h. The average contact with the facial mucosa (frequency/person/2 h) was 55.6 ± 42.2, of which the mouth (25.4 ± 23.9), the nose (20.4 ± 24.5), and the eye (9.8 ± 11.7) were the most frequent contacts, in that order. The average contact duration (sec/person/2 h) with the facial mucosa was 232.6 ± 169.7, of which the mouth (145.2 ± 150), the nose (57.6 ± 62.2), and the eyes (29.7 ± 42.3) were the longest in that order. The density distribution of the frequency and duration of mucosal contact was wider in boys than in girls. The mucosal and non-mucosal contact frequencies were significantly higher in boys (*p* = 0.027 and *p* = 0.030, respectively). Conclusion: Children’s HFC frequency and duration were highest for the mouth, nose, and eyes. Boys tended to have a higher contact frequency than girls for both mucous and non-mucous HFC.

## 1. Introduction

The most common mode of person-to-person spread of infection is contact with contaminated hands [1]. Contaminated hands can also transmit infection within a person from one part of the body to another, i.e., self-inoculation [2]; in particular, hand-to-face contact (HFC) is one of the major routes for self-inoculation of infectious pathogens [3]. When someone touches their own mucous membranes, such as the eyes, nose, or mouth with contaminated hands, self-inoculation of infectious pathogens can frequently occur [4]. 

Self-inoculation via HFC in individuals leads to further opportunities for transmission of infections, such as respiratory infections in adults [5], contact type infections in infants [2] and zoonotic disease in children [6]. In the current coronavirus disease (COVID-19) pandemic, HFC has been reported as a possible source of COVID-19 transmission [7,8].

HFC occurs frequently in daily life with 50.06 (±47) facial touches per hour, and 68.7 (±27) touches per hour within the T-zone (eyes, nose, mouth, chin) [9]. To prevent the spread of infection by habitual HFC, habit-reversal training (HRT) has been advised as a possible public health intervention for reducing HFC behaviors that increase the risk of infection [10]. 

For children, hand-to-mouth activities have been studied as a major route of ingestion of toxic materials, such as soil, dust, chemicals, or diarrhea-inducing materials [11,12]. However, to the best of our knowledge, few studies on HFC that include the eyes, nose, and mouth of children have been reported, despite the frequent occurrence of HFC and its importance for self-inoculation as one of the main routes of respiratory viruses and other pathogens [13]. 

Moreover, few studies have described children’s HFC in terms of hand-to-eye, -nose, or -mouth contact in the Republic of Korea (hereafter, Korea). As many children spend their time in childcare facilities [14], there are concerns about increased risk of infection [15]. Therefore, it is necessary to understand the characteristics of HFC in children to develop infection prevention strategies addressing HFC as a means of self-inoculation of pathogens.

The aims of this study were to characterize HFC (to the eyes, nose, mouth, and other areas of the face) in children and to analyze the factors that affect the HFC behaviors of preschoolers under six years of age in childcare facilities in Korea.

## 2. Materials and Methods

### 2.1. Study Participants

The study participants were preschool children aged between 13 and 84 months, attending a group-care facility, such as a daycare center, recruited using convenience sampling and snowballing methods. Since children are underage and vulnerable, they were enrolled only from childcare centers where the directors and the teacher in charge voluntarily consented and agreed to participate in the study; the parents of the children in childcare centers voluntarily provided consent to participate after being informed of the study’s purpose and methods. Thirty children (10 each from two childcare centers (A, B) in Seoul and 10 from one childcare center (C) in Gyeonggi Province) were enrolled in the study. Childcare centers A and C participated based on the researchers’ local knowledge, and center B participated using snowballing methods founded on center A. The participants were children without physical or mental disabilities.

### 2.2. Videotaping and Data Collection

This study was an observational study involving video recording. Videotaping was performed in the presence of a member of the research team after notice was provided of the videotaping process. The taping was recorded as video only, and the audio was silent. Two video cameras were installed to secure sufficient data for analysis. The video equipment supported high-quality auto-focusing that maintained focus without losing details of the participants’ motion [16]. Videotaping was undertaken (10 children/childcare center/2 h) on November 23 at childcare center C, and on 14 and 16 December 2018, at childcare centers A and B, respectively. Contact data were recorded and videotaping occurred at the time of the day when activities were at their maximum.

Data for contact frequency (CF) and contact duration (CD) of HFC behaviors were recorded by observing the videotape for each participant in standardized Excel format. The read time to check each CD was recorded in the format of start minute, end minute, start second, and end second (referring to the timestamp at the bottom of the video recording) to avoid missing time. We checked video playback to avoid loss of data. Contact sites were observed and HFCs were classified according to whether they involved contact with the mucous membrane. Data were collected from 30 November 2018 to 7 January 2019.

To ensure the accuracy and reliability of the image reading, two observers, who could cross-verify, were secured, and training was conducted three times. To reduce intra-individual read errors, each reader re-read 4% of the data entered and checked the contact area, CF, and CD to ensure that they were more than 90% consistent. Minimization of interpersonal reading errors was confirmed with approximately 10% to 90%, or greater, agreement of randomly selected reads by other observers.

### 2.3. Definition of Terms

Mucosal contact was defined as direct contact involving the eyes, nose, and mouth, while non-mucosal contact was defined as direct contact involving the skin of the head, forehead, chin, cheeks, and ears.

### 2.4. Data Analysis

Descriptive statistics, including for both continuous and categorical variables, are presented as mean ± standard deviation (SD), range (min, max), percentile (0.05, 0.25, 0.5, 0.75, 0.95), and frequencies and percentages (%), respectively. Differences in children’s gender and age were investigated as factors influencing children’s HFCs behavior. The CF and CD according to the gender of the participants were compared by visualizing the density distribution. To compare the differences in HFC by gender and age of children, a t-test was used with statistical significance assumed at α = 0.05; boxplots using ggplot2 were created. Data analyses were performed using R 3.3.3 (version 3.15) for Windows (R Foundation for Statistical Computing, Vienna, Austria).

### 2.5. Ethical Considerations

This study was approved by the institutional review board (IRB 1040173-201809-HR-026-04). In addition, informed consent was obtained from parents who voluntarily participated in the study. Before obtaining voluntary consent, the participants’ parents were informed of the study objectives, methods, expected benefits, risks of participation, and the freedom to withdraw from participation and to withdraw consent at any time. Written consent was obtained from the parents of all participants before and after video recording.

## 3. Results

### 3.1. General Characteristics of the Participants and Their Parents

In total, 70.0% of the children in the sample were aged 37–60 months (mean = 53.1 months) and 56.7% were male. (Table 1).

### 3.2. Descriptive Statistics of Frequency and Duration for Hand-to-Face Contact

In 2 h, 2719 cases of HFC for 30 participants were observed. The average contact (frequency/person/2 h) was 55.6 ± 42.2 with the facial mucosa, of which the mouth (25.4 ± 23.9), nose (20.4 ± 24.5), and eye (9.8 ± 11.7) were the most frequent contacts, in that order (Appendix A) (Figure 1a and Figure 2). On average, the total CD (sec/person/2 h) with the facial mucosa per person was 232.6 ± 169.7; of these, contacts with the mouth (145.2 ± 150), nose (57.6 ± 62.2), and eyes (29.7 ± 42.3) were the longest, in that order (Appendix A) (Figure 1b).

### 3.3. Density Distribution of Frequency and Duration for Hand-to-Face Contact by Gender

The density distribution of frequency and duration of mucosal contact, including the eyes, nose, and mouth, showed a wider distribution in boys than in girls (Figure 3).

### 3.4. Differences in CF and CD by the Child’s Age and Gender

There were no statistically significant differences in the frequency and duration of mucosal contact according to age. The mucosal and non-mucosal contact frequencies were significantly different according to gender (*p* = 0.027 and *p* = 0.030, respectively). Boys tended to touch their mouth, eyes, and nose more frequently than girls, and more than other areas of the face, i.e., non-mucous areas. However, there were no differences in contact duration between mucous and non-mucous contacts (Figure 4).

## 4. Discussion

The average HFC frequencies in this study were lower than that reported for specific touch of the T-zone in a previous review in adults (50.06 (±47), 68.7 (±27)/h) [9]. The average HFC frequencies in this study were higher than hand-to-mouth behavior, ranging from 6.7 to 28.0 contacts/hour for children aged less than 11 years [12]. The average HFC frequencies in this study were lower than mouthing behavior, with 81 +/− 7 events/h for children aged ≤24 months, and higher than for children >24 months, who exhibited the lowest frequency of four mouthing behaviors with 42 +/− 4 events/h [5,17]. The average HFC frequencies were lower than those for mouthing with 43–72/h [18].

Median HFC frequencies in this study were higher than those observed in a study of hand-to-mouth contact in children aged 3 to <6 years old which reported 10/hour [11], and of hand-to-mouth contact in children aged 7 to 35 months, with 8.91/hour reported [19].

The HFC frequencies in this study were higher than those obtained for adults in previous studies [13,20]. HFC frequencies in this study were highest in the mouth, followed by the nose and the eye, consistent with the results of a previous study, which found significantly higher HFC for the mouth and nose than adults [13]. The average frequency of hand-to-mouthing was reported as 6.7 to 28.0 contacts/hour and decreased with age, with the highest values observed for the 3- to 6-month age group [12]. HFC frequencies were higher in younger children aged ≤24 months than in older children [5,17], and were significantly negatively correlated with age [11,21,22]. In contrast to previous studies [11,12,18,19,21,22], HFC frequencies were not significantly different between the >48 and ≤48 months age groups. This may be related to the fact that the participants in this study were older than those in previous studies [10,19,20]. However, this study precisely determined the HFC frequency and duration according to the area touched of both the mucosal and non-mucosal areas of the face. This is useful in estimating the risk of self-inoculation of infections. In addition, HFC frequencies in preschoolers were highest for the mouth, followed by the nose and the eye; therefore, the mouth may be a major route of infectious agents for them, too. Avoiding HFC habits, frequent attention to hand hygiene and implementing strict hygienic procedures for every toy or instrument in childcare centers must be emphasized to a greater degree to prevent the spread of infection, toxic substances, and oro-fecal diseases [11,12].

The median HFC duration in this study was higher than the median indoor hand-to-mouth hourly contact duration, i.e., 0.34 min/h [19]. The authors could not find other studies on HFC durations, so, a limited number of sources exist that may be compared with the results of this study.

Interestingly, boys showed lower but wider density of HFC, in terms of both frequency and duration, than girls for contact with mucous membranes, and higher CF than girls for both mucous and non-mucous HFC in this study, which has not been addressed in previous studies. To the best of our knowledge, only one meta-analysis has shown that gender is not important for hand-to-mouth frequency [12]. However, this study showed that boys contacted both mucous and non-mucous HFC more frequently than girls. This could be the result of behavioral differences by gender. These differences in HFC by gender may be associated with the observed higher incidence of respiratory infectious disease in boys with severe RSV infection, with a male-to-female ratio of 1.4 [23]. Boys generally showed higher levels of activity [24] and riskier behaviors than girls, so they experience more injuries as pedestrians than girls [25]. Boys showed higher HFC than girls in this study; this may be related to the characteristics of boys identified in previous studies [23,24,25]. Boys can also be exposed to a high level of risk of infections by HFC. Boys should be encouraged to avoid and prevent HFC as part of safety education and trained in good habits of hand hygiene to prevent the transmission of infections.

HFC, especially to mucus membranes, the eyes, nose, and mouth, can be a route for self-inoculation of infectious pathogens [4,5]. Therefore, the frequency and duration of HFC can be useful indices for estimating the risk of respiratory pathogens. During the period of COVID-19, it is strongly recommended to avoid touching the face, eyes, nose, or mouth with unwashed hands to prevent the spread of COVID-19 [26]. During the COVID-19 pandemic [27], the risk of COVID-19 outbreaks in childcare facilities cannot be ignored [28]. HFC can be useful for estimating microbial exposure and pathogen transmission [13]. Therefore, the findings of this study will be useful in developing preventive measures for children to protect against self-inoculation of infectious pathogens via HFC. This study has some limitations with respect to the populations sampled and the locations used. As the results were obtained via convenience sampling, they cannot be generalized to the broader child population because of potential bias that may have resulted in under-representation of subgroups in the sample in comparison to the broader population of children. Further studies need to be conducted with children’s groups of various ages and in various locations, including rural areas. However, this study provides useful evidence about HFC in children as a means of infection transmission. Children must be repeatedly reminded about hand hygiene, and childcare centers must take care to frequently disinfect environmental surfaces in childcare centers to avoid HFC and to prevent the spread of infections.

## 5. Conclusions

The frequency and duration of HFC in children was found to be in the order of mouth, nose, and then eyes. Among the mucous membranes, the mouth can be a major route of self-inoculation in children. Strict hand hygiene and disinfection of all surfaces in the childcare environment, especially play equipment, must be emphasized. Boys showed more frequent HFC and may be exposed to higher risks of infection transmission by HFC. Boys should be encouraged to avoid and prevent HFC as part of safety education and trained in good habits of hand hygiene to prevent the transmission of infections.

## Figures and Tables

**Figure 1 ijerph-19-13282-f001:**
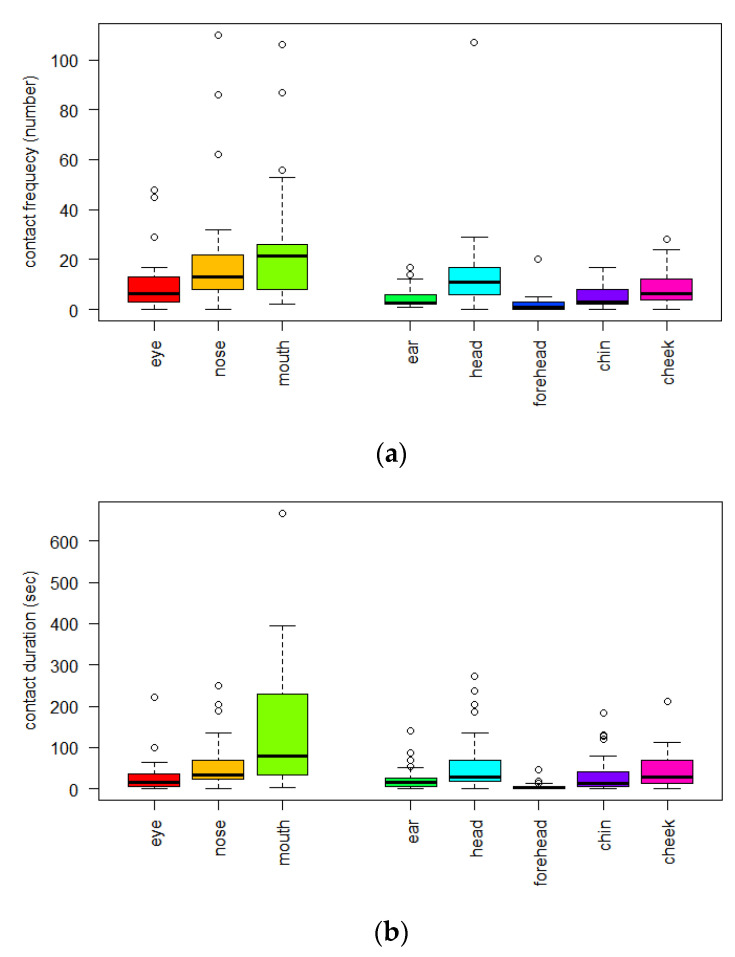
(**a**) Hand-to-face contact frequency (*n* = 30, 2 h observation time, frequency/person/2 h); (**b**) Hand-to-face contact duration (*n* = 30, 2 h observation time, sec/person/2 h).

**Figure 2 ijerph-19-13282-f002:**
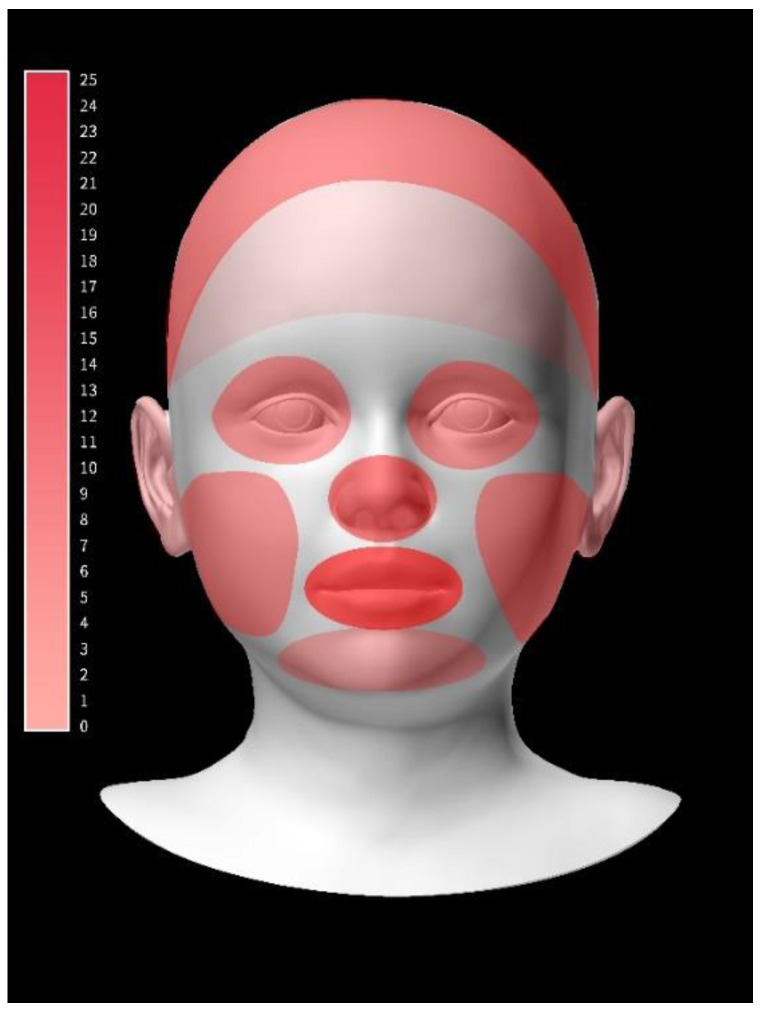
Visualizing frequency of hand-to-face contact (number/person/2 h). The color bar expresses the contact frequency from 0 to 25. The darker the color, the more frequent the contact.

**Figure 3 ijerph-19-13282-f003:**
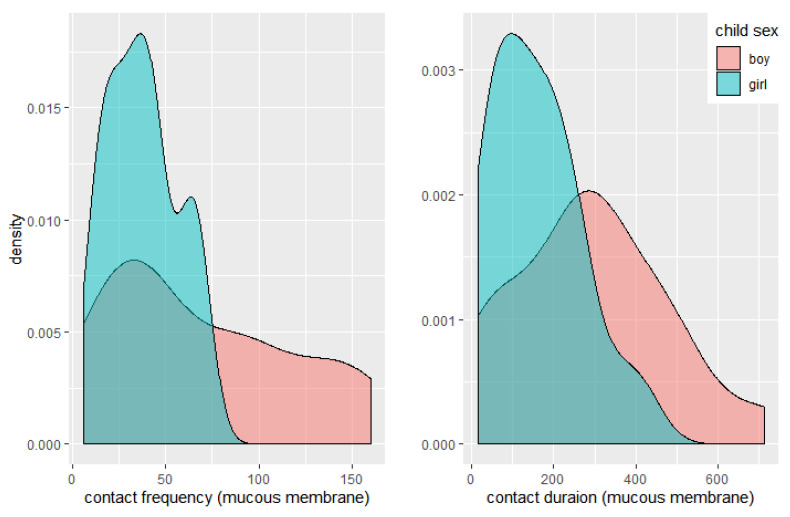
Density distribution of frequency and duration in hand-to-face contact by gender. Density of contact frequency = number/contact frequency-group width; Density of contact duration = number/contact duration-group width.

**Figure 4 ijerph-19-13282-f004:**
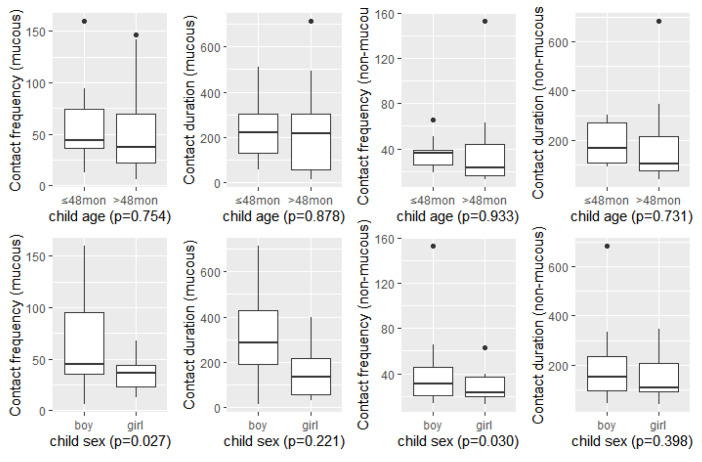
Differences in contact frequency and contact duration by child age and gender. Single variance analysis (*n* = 30, 2 h observation time).

**Table 1 ijerph-19-13282-t001:** General characteristics of the participants and their parents.

Variable	Classification	Number of Participants *n* (%) or Mean ± SD
		30
Age of children (month)		53.1 ± 13.7
	13–36	3 (10.0)
	37–60	21 (70.0)
	61–84	6 (20.0)
Gender of child		
	Boy	17 (56.7)
	Girl	13 (43.3)
Number of people living together		3.7 ± 0.7
	≤3	11 (36.7)
	>4	19 (63.3)
Residential area		
	Seoul	16 (53.3)
	Gyeonggi-do	14 (46.7)
Child’s parents Age (year)		37.0 ± 3.7
	25–39	21 (70.0)
	40–49	9 (30.0)
Education		
	High school	2 (6.7)
	College/University	22 (73.3)
	Graduate school	6 (20.0)
Employment status		
	No	18 (60.0)
	Yes	12 (40.0)
Type of work		
	Permanent employee	8 (26.7)
	Part-time work	4 (13.3)
	Unemployed, including housewife	18 (60.0)
Household income (won)		
	≤4 million	15 (50.0)
	>4 million	15 (50.0)

## Data Availability

Data sharing not applicable.

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
