# Peer review of "Hand-to-Face Contact of Preschoolers during Indoor Activities in Childcare Facilities in the Republic of Korea"

_ijerph, 2022, doi:10.3390/ijerph192013282_

Round 1

Reviewer 1 Report

Overall – General proofing for English grammar.

Abstract -

Line 9 – Hand-to-face contact shouldn’t be capitalized.

Line 19 – Should it read “showed a wider range in boys…”?

Introduction –

Line 27 – Is “common” maybe a more appropriate word in place of “frequent”?

Line 37 – Should this read “with an average of 50.06 facial touches per hour, and 68.7 touches per hour within the T-zone…”?

Line 45 – I think this should read “few studies of HFC including the eyes…. have been reported”.

More detail should be provided on the types/severity of illnesses spread through HFC – especially for children.  Potential additional areas include the types of illnesses spread in other types of communal areas, such as senior center, schools, etc.

Methods –

How were the childcare centers identified?

What was the purpose of determining gender differences in HFC behaviors?

Were the activities in which the children were involved similar across all childcare centers?  How did researchers ensure that activities in which the children were involved were similar? For example, if one group of children was playing inside, while another group was playing outside, that might impact results.  Additionally, if one or more groups were having snack or eating lunch during the recording, and others were not, that might also impact the overall results.

Results -

Please describe in greater detail the general characteristics – what is cohabitation number? What is meant by occupation yes/no – is this whether or not they were employed?

Figure 1 – Is there a way to ensure that figure 1 is provided in a more cohesive manner? Currently it spans 2 pages.

Line 130 - Please provide a definition for “density distribution” and why this might be important.

Discussion -

Authors should provide additional rationale as to the importance of these findings.  They briefly touch on COVID, but what are the overarching public health implications of the study?  How can these results be used to develop interventions to address these behaviors?  Why might it be important that there are gender differences in these behaviors?

Conclusions -

Similar comments provided in the discussion section above.  Authors should provide a statement as to the importance of the study, and why it might be important to understand the differences in these behaviors by gender.

Reviewer 2 Report

The article is interesting but it should be more in-depth, namely in its theoretical framework.

It is not possible to understand why it was carried out in two different residential areas and if any relationship was noticed in relation to this variable and why these two locations were chosen (urban and rural areas?)

Reviewer 3 Report

Dear Authors,

The article entitled "Hand-to-Face Contact of Preschoolers during Indoor Activities in Childcare Facilities in the Republic of Korea" underlines the fact that these movement are very frequent in case of children, and this may represent a risk factor for infection transmission.

Here are some observations that I made upon the article:

In the Abstract section: line 20 - please consider "gender" instead of "sex" (and throughout the entire paper as well).

In the Introduction section: more information is necessary, especially concerning the relationship between HFC and pathogens transmission.

In the Materials and Methods section:

-how was sample size calculated?

-the inclusion/exclusion criteria should be specified

-the test/null hypothesis should be formulated.

In the Discussion section:

-lines 147-148 - please rephrase, as the expression is unclear

-the limitations of the study should be discussed.

The References should follow the recommended style for MDPI journals.

The most important observation is that the Authors should provide a clearer image of the relationship between the results of the study and infection transmission, and thus present the practical significance of the results. One way would be to make the connection between the study results and other aspects important for pathogens transmission.

Round 2

Reviewer 1 Report

The authors have adequately addressed my comments and concerns.  Some additional English language proofing is required.

Author Response

Dear, reviewer

We are grateful for the careful review of our paper.

The certificate for English editing is attached.

Reviewer 3 Report

Dear authors,

Thank you for considering most of my recommendations for your article.

Some observations still remain valid:

-the term "gender" is used in scientific literature to designate differences between males and females, irrespective of age, as the previously used term "sex" might have sexual connotations

-"convenience sampling" is a method for sample selection, not for sample size calculation. Anyway, the inconveniences of convenience sampling should be mentioned: The results of the convenience sampling cannot be generalized to the target population because of the potential bias of the sampling technique due to the under-representation of subgroups in the sample in comparison to the population of interest. The bias of the sample cannot be measured. Therefore, inferences based on convenience sampling should be made only about the sample itself. Convenience sampling is characterized with insufficienpower to identify differences of population subgroups:

Bornstein, Marc H.; Jager, Justin; Putnick, Diane L. (28 April 2017). "Sampling in Developmental Science: Situations, Shortcomings, Solutions, and Standards"Developmental Review33 (4): 357–370. doi:10.1016/j.dr.2013.08.003

Bornstein, Marc H.; Jager, Justin; Putnick, Diane L. (28 April 2017). "Sampling in Developmental Science: Situations, Shortcomings, Solutions, and Standards"Developmental Review33 (4): 357–370. doi:10.1016/j.dr.2013.08.003

The article needs English correction.

Author Response

Dear, reviewer

We are grateful for the careful review of our paper. We have addressed the comments raised, and the amendments are highlighted in red in the revised manuscript. Point-by-point responses of correction to your’ comments are listed below this table.

comment

correction

-the term "gender" is used in scientific literature to designate differences between males and females, irrespective of age, as the previously used term "sex" might have sexual connotations

Corrected the term to "gender".

-"convenience sampling" is a method for sample selection, not for sample size calculation. Anyway, the inconveniences of convenience sampling should be mentioned: The results of the convenience sampling cannot be generalized to the target population because of the potential bias of the sampling technique due to the under-representation of subgroups in the sample in comparison to the population of interest. The bias of the sample cannot be measured. Therefore, inferences based on convenience sampling should be made only about the sample itself. Convenience sampling is characterized with insufficient power to identify differences of population subgroups:

Bornstein, Marc H.; Jager, Justin; Putnick, Diane L. (28 April 2017). "Sampling in Developmental Science: Situations, Shortcomings, Solutions, and Standards". Developmental Review33 (4): 357–370. doi:10.1016/j.dr.2013.08.003

In this study, based on the concept of the central limit theorem, a minimum sample size of 30 or more was secured as much as possible. but this number may not be appropriate.

We fully agree with your comments on the limitations of sample size calculations and generalization of the results of this study. Therefore, we added limitations of this study based on the recommendations in the discussion section.

Line 210-213

This study has some limitations regarding study populations and places for generalization. The results of this study from convenience sampling cannot be generalized to the children population because of the potential bias of the sampling due to the under-representation of subgroups in the sample in comparison to the population of children. Further studies will be needed to conduct with children’s groups of various ages and in various locations such as rural areas. 

The article needs English correction. The certificate for English editing is attached.
